# Efficient Direct and Limited Environmental Transmission of SARS-CoV-2 Lineage B.1.22 in Domestic Cats

Nora M. Gerhards,[a] Jose L. Gonzales,[a] Sandra Vreman,[b] Lars Ravesloot,[b] Judith M. A. van den Brand,[c] Harmen P. Doekes,[d] Herman F. Egberink,[e] Arjan Stegeman,[f] Nadia Oreshkova,[g] Wim H. M. van der Poel,[g] Mart C. M. de Jong[h]

[a]Department of Bioinformatics, Epidemiology and Animal Models, Wageningen Bioveterinary Research, Lelystad, the Netherlands

[b]Department of Bacteriology, Host-Pathogen Interactions and Diagnostic Development, Wageningen Bioveterinary Research, Lelystad, the Netherlands

[c]Division of Pathology, Faculty of Veterinary Medicine, Utrecht University, Utrecht, the Netherlands

[d]Animal Breeding and Genomics, Wageningen University and Research, Wageningen, the Netherlands

[e]Division Infectious Diseases and Immunology, Section Virology, Faculty of Veterinary Medicine, Utrecht University, Utrecht, the Netherlands

[f]Department of Population Health Sciences, Veterinary Epidemiology, Faculty of Veterinary Medicine, Utrecht University, Utrecht, the Netherlands

[g]Department of Virology and Molecular Biology, Wageningen Bioveterinary Research, Lelystad, the Netherlands

[h]Quantitative Veterinary Epidemiology, Wageningen University, Wageningen, the Netherlands

**ABSTRACT** The susceptibility of domestic cats to infection with SARS-CoV-2 has been demonstrated by several experimental studies and field observations. We performed an extensive study to further characterize the transmission of SARS-CoV-2 between cats, through both direct and indirect contact. To that end, we estimated the transmission rate parameter and the decay parameter for infectivity in the environment. Using four groups of pair-transmission experiment, all donor (inoculated) cats became infected, shed virus, and seroconverted, while three out of four direct contact cats got infected, shed virus, and two of those seroconverted. One out of eight cats exposed to a SARS-CoV-2-contaminated environment became infected but did not seroconvert. Statistical analysis of the transmission data gives a reproduction number $R_0$ of 2.18 (95% CI = 0.92 to 4.08), a transmission rate parameter $\beta$ of 0.23 day$^{-1}$ (95% CI = 0.06 to 0.54), and a virus decay rate parameter $\mu$ of 2.73 day$^{-1}$ (95% CI = 0.77 to 15.82). These data indicate that transmission between cats is efficient and can be sustained ($R_0 > 1$), however, the infectiousness of a contaminated environment decays rapidly (mean duration of infectiousness 1/2.73 days). Despite this, infections of cats via exposure to a SARS-CoV-2-contaminated environment cannot be discounted if cats are exposed shortly after contamination.

**IMPORTANCE** This article provides additional insight into the risk of infection that could arise from cats infected with SARS-CoV-2 by using epidemiological models to determine transmission parameters. Considering that transmission parameters are not always provided in the literature describing transmission experiments in animals, we demonstrate that mathematical analysis of experimental data is crucial to estimate the likelihood of transmission. This article is also relevant to animal health professionals and authorities involved in risk assessments for zoonotic spill-overs of SARS-CoV-2. Last but not least, the mathematical models to calculate transmission parameters are applicable to analyze the experimental transmission of other pathogens between animals.

**KEYWORDS** cats, SARS-CoV-2, direct transmission, indirect transmission, COVID-19, susceptibility

Address correspondence to Nora M. Gerhards, nora.gerhards@wur.nl, Wim H. M. van der Poel, wim.vanderpoel@wur.nl, or Mart C. M. de Jong, mart.dejong@wur.nl.

The authors declare no conflict of interest.

*[This article was published on 24 May 2023 with missing cities in the affiliations. The affiliations were corrected in the current version, posted on 30 May 2023.]*

Since the first cases of atypical pneumonia due to SARS-CoV-2 were identified in December 2019, a large number of animal studies have been performed to assess which species are susceptible to SARS-CoV-2 infection. Next to susceptibility to SARS-

CoV-2, it is relevant to determine which species develop clinical disease and which species can transmit the virus. In several studies, animals were experimentally inoculated with high doses of SARS-CoV-2 to demonstrate species susceptibility, including ferrets, cats, hamsters, nonhuman primates, and rabbits (reviewed by Jo et al. [1]). Next to these experimental infections, a number of animals were shown to be infected naturally by contact with infected humans, such as minks (2), dogs (3), and both domestic and big cats (4, 5).

It is relevant to assess the risk of SARS-CoV-2 transmission between animals and humans in connection with species susceptibility to infection and/or disease. This is of particular concern for animal species that live in close contact with humans. These animals can be a reservoir and/or an intermediate host for humans. There are a few experimental studies that address transmission between cats, which show that infected cats can pass on the virus by aerosol or contact transmission to other cats (6–9). A commonality of these studies is an inoculation dose above $10^5$ PFU/TCID$_{50}$, housing of animals in smaller isolators/cages and frequent sampling under anesthesia. A systematic review of these experimental data combined with household data confirmed that sustained direct contact transmission between cats is probable ($R_0 > 1$) and that the risk of indirect transmission via aerosols is possible, although less likely than direct contact transmission (10).

The risk of transmission via exposure to contaminated surfaces (environment) has not been studied yet. Assessing this risk is relevant because susceptible cats could be exposed to contaminated surfaces during visits to pet shelters or veterinary clinics if proper hygienic measures are not implemented. In addition, successful infection of cats exposed to an environment contaminated with SARS-CoV-2 by other cats could be an indicator of the potential risk for cat-to-human transmission via a contaminated environment. While the transmission of SARS-CoV-2 from humans to animals is well-documented, reports of transmission from animals to humans remain limited. To date, mink-to-human transmission has been confirmed (11), and recent events have highlighted potential transmission from infected hamsters to humans in a pet-shop in Hong Kong (12). Moreover, a recent report describing a possible cat-to-human transmission of SARS-CoV-2 highlights the possibility of this transmission pathway (13), which is relevant given the proximity between humans and their pets.

To assess the potential risk of cats for spreading SARS-CoV-2 by contaminated environment and to provide further evidence of direct contact transmission, we performed a transmission experiment in cats after infection with SARS-CoV-2. We confirm that direct contact transmission of SARS-CoV-2 is efficient between cats and that the infectiousness of a contaminated environment decays rapidly, thereby limiting the efficiency of environmental transmission alone. Nevertheless, the indirect environmental transmission route may still be possible.

## RESULTS

The study design is explained in detail in the Methods section and summarized in Fig. 1. In brief, a total of 16 cats were divided into four replicate groups (group 1, 2, 3 and 4) consisting of either four male or four female cats per group. Each group was divided in two subgroups, which were housed in two separate pens A and B: cats X.1 and X.2 were housed together in pen A from day 1 (D1) until D6 and in pen B from D7 to 23, and cats X.3 and X.4 were housed together in pen B from D0 to 6 and in pen A from D7 to 23, where X = experimental group. All four cats identified by X.1 were inoculated on D0 and all cats identified by X.2 were housed together with X.1 except on the day of inoculation of cats X.1 (D0). On D6, cats X.1 and X.2 were removed from pen A and cats X.3 and X.4 were placed in (contaminated) pen A; cats X.1 and X.2 were then placed in (clean) pen B in which cats X.3 and X.4 had been housed until then. Thus, four independently housed pairs of cats were used to assess direct transmission (cats X.1 and X.2) and contamination of the environment (pen) where new naive cats (two per contaminated pen: X.3 and X.4) were introduced following the removal of the

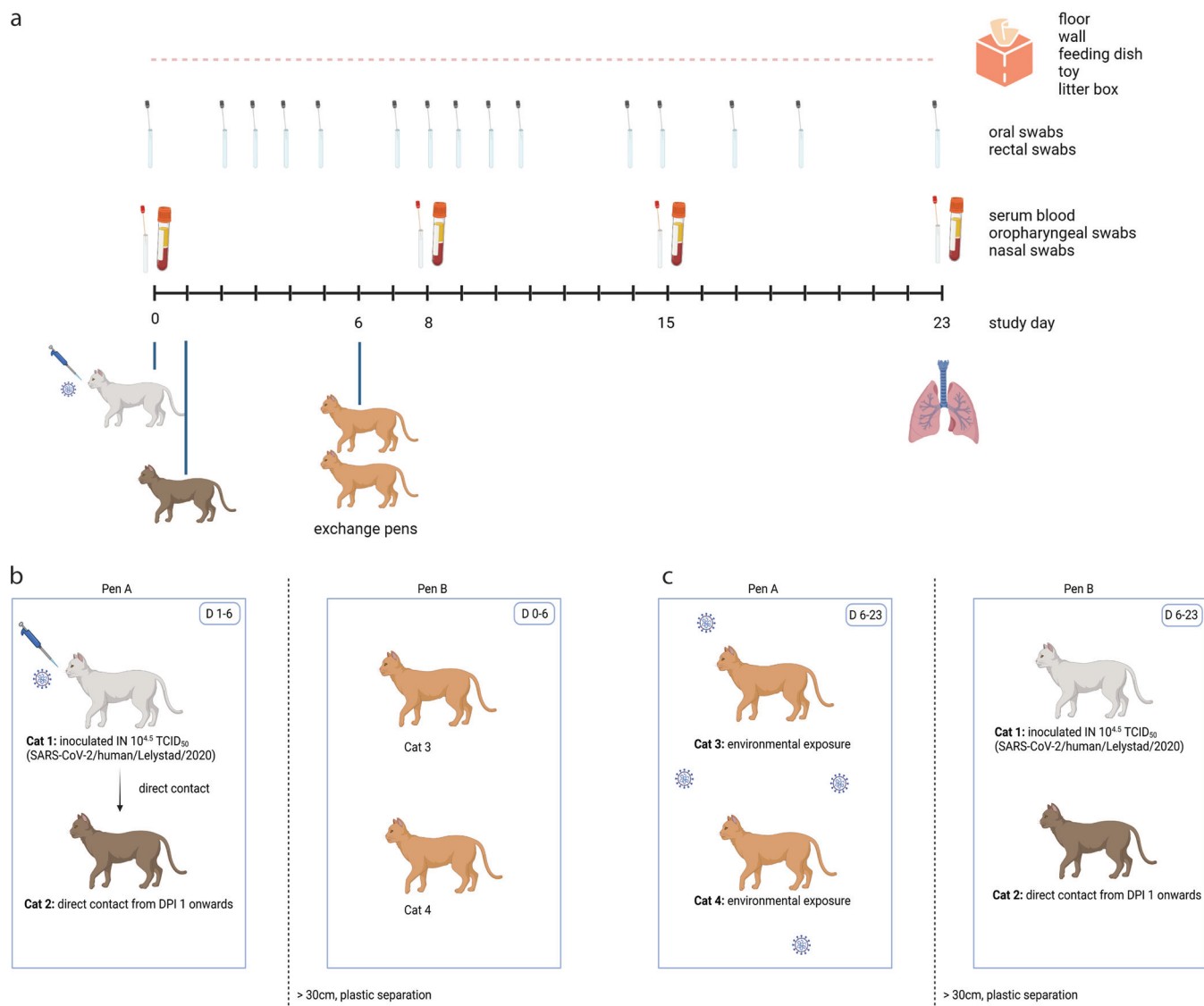

**FIG 1** Experimental design of one out of four replicate groups. (a) Schematic timeline. One cat was inoculated intranasally on D0 and was brought into contact with a naive cat on D1 in pen A. On D6, these two cats were exchanged with two other naive cats. Blood samples, oropharyngeal swabs and nasal swabs were collected under general anesthesia on D0, D8, D15, and upon euthanasia. Oral and rectal swabs were collected more frequently without anesthesia. Environmental samples were collected daily, and all cats were euthanized on D23 except of one cat who died before the end of the study by a cause not related to SARS-CoV-2. (b) Until D6, cat 1 (inoculated donor cat) and cat 2 (direct contact recipient cat) were housed in pen A and contaminated the environment. Cat 3 and cat 4 were housed in pen B, which was separated from pen A by a plastic separator and space. (c) From D6 onwards, cats 3 and 4 were housed in contaminated pen A. Cats 1 and 2 were housed in pen B.

pair of cats used to assess direct transmission. The sample size was calculated based on published data (10). Of the direct transmission pairs, the four donor cats became infected following inoculation, and three of those transmitted infection to their corresponding contact cats. Of the eight cats exposed to the contaminated pens, only one got infected. Below, we provide the detailed results of the observed infection characteristics and the quantitative assessment of transmission.

**Clinical signs and pathology.** Two of the four inoculated cats occasionally showed mild serous nasal discharge. All 12 cats that had direct contact or indirect contact with an inoculated cat remained without clinical signs. The body weights of all 16 cats remained constant. A video-based analysis of activity of the direct transmission pairs revealed no changes in activity post inoculation compared to before inoculation (baseline measurement), indicating that inoculated and direct contact cats displayed a similar activity pattern regardless of SARS-CoV-2 inoculation/direct exposure (Supplemental Material 1). One cat

from group 4 (cat 4.3) died on D18 and was necropsied the same day. It was confirmed that the death was unrelated to SARS-CoV-2 infection. All other cats were euthanized on D23. Upon necropsy, no lesions were observed in gross pathology, and there were no substantial differences in relative lung weights between the animals, as expected 2 to 3 weeks post-SARS-CoV-2 exposure (data not shown). Lung tissue showed no SARS-CoV-2-related histopathology. However, mild lung changes, such as lymphoplasmacytic bronchoadenitis, bronchus-associated lymphoid tissue (BALT) hyperplasia, infiltrates of macrophages, and few neutrophils in alveolar lumina, were observed in inoculated animals, and in direct and indirect contact animals (data not shown) suggesting these changes were nonspecific. Yet, transient lung pathology at earlier time points that were resolved until day of necropsy, cannot be excluded. Other evaluated organs (nasal conchae, trachea, tracheobronchial lymph node, duodenum, ileum, colon, mesenterial lymph node, and pancreas) showed also no substantial histopathological findings. No viral antigen could be detected by immunohistochemistry in any of the investigated tissues (data not shown).

**Viral RNA load in swabs and organs.** Oral and rectal swabs were taken frequently, while nasal and oropharyngeal swabs were only sampled on a few time points (D0, D8, D15, and D23). Swabs were analyzed by both total E-gene PCR according to Corman et al. (14), and by subgenomic PCR (sgPCR) according to Wolfel et al. (15). No virus isolations were attempted, because many swab samples had high Ct values ($>$30) from which virus can only occasionally be isolated (16, 17).

SARS-CoV-2 RNA was detected by both total E-gene PCR as well as sgPCR in multiple swabs of all inoculated cats (X.1) from D2 until the end of the study (Fig. 2a). In swabs of all four direct contact cats (X.2), SARS-CoV-2 total E-gene RNA was detected, while only three cats (1.2, 2.2, and 4.2) also had at least one positive sample in the sgPCR. Of the indirect contact cats (X.3 and X.4) exposed to contaminated environment, total E-gene RNA was detected in swabs in groups 1, 3, and 4, while in group 2, both cats remained negative for SARS-CoV-2 RNA (Fig. 2b). Of all swabs collected from all indirect contact cats, only one sample tested positive in the sgPCR (oral swab of cat 1.3 on D7).

After necropsy, respiratory organs (nasal conchae, trachea, and lungs) of all cats were analyzed by total E-gene PCR. All samples were PCR-negative except of nasal conchae of cats 1.1, 2.1, and 4.1 (three of the four inoculated cats). It is worthy of note that these samples were collected 23 days postinoculation.

**Serology and immunology.** Sera and heparinized blood samples were collected on D0, D8, D15, and D23 from all cats. Virus neutralizing antibodies were detected in all inoculated cats (X.1) on D8, D15, and D23 with neutralizing titers ranging from 90 to 810 (Fig. 3a). Also, anti-S1 antibodies were detected in all inoculated cats by ELISA (Fig. 3b). Two out of the four direct contact cats (1.2 and 2.2) developed neutralizing (titer range 5.7 to 155) and anti-S1 antibodies, while the other two direct contact cats (3.2 and 4.2) did not seroconvert. None of the indirect contact cats developed a serological response.

To evaluate memory T-cell interferon-$\gamma$ production, peripheral blood mononuclear cells (PBMCs) were stimulated for 48 h with a SARS-CoV-2 spike (S), nucleoprotein (NP), or a SARS-CoV-2 spike protein peptide pool, and were subsequently analyzed with an ELISpot assay (Fig. 3c). Compared to medium stimulation, a clear interferon-$\gamma$ response (increased number of spots) was measured in all four inoculated cats and two direct contact cats (1.2 and 2.2). This indicates that these cats were able to mount a specific cellular immune response on D23 (inoculated cats) or between D17 to 22 (direct contact cats). In none of the indirect exposure cats, a memory T-cell interferon-$\gamma$ response was observed (X.3 and X.4). For cats 3.2 and 4.3, we obtained insufficient PBMCs to perform an ELISpot assay.

**Environmental contamination.** Environmental samples (floor, wall, litter box, feeding tray, and toy) were collected daily from all pens A, and analyzed by total E-gene PCR. Viral RNA could be detected in all types of samples until the end of the study (Fig. 3d). There was variation between the groups, among which the highest median genome copy numbers in the different sampled surfaces were found in group 4, and

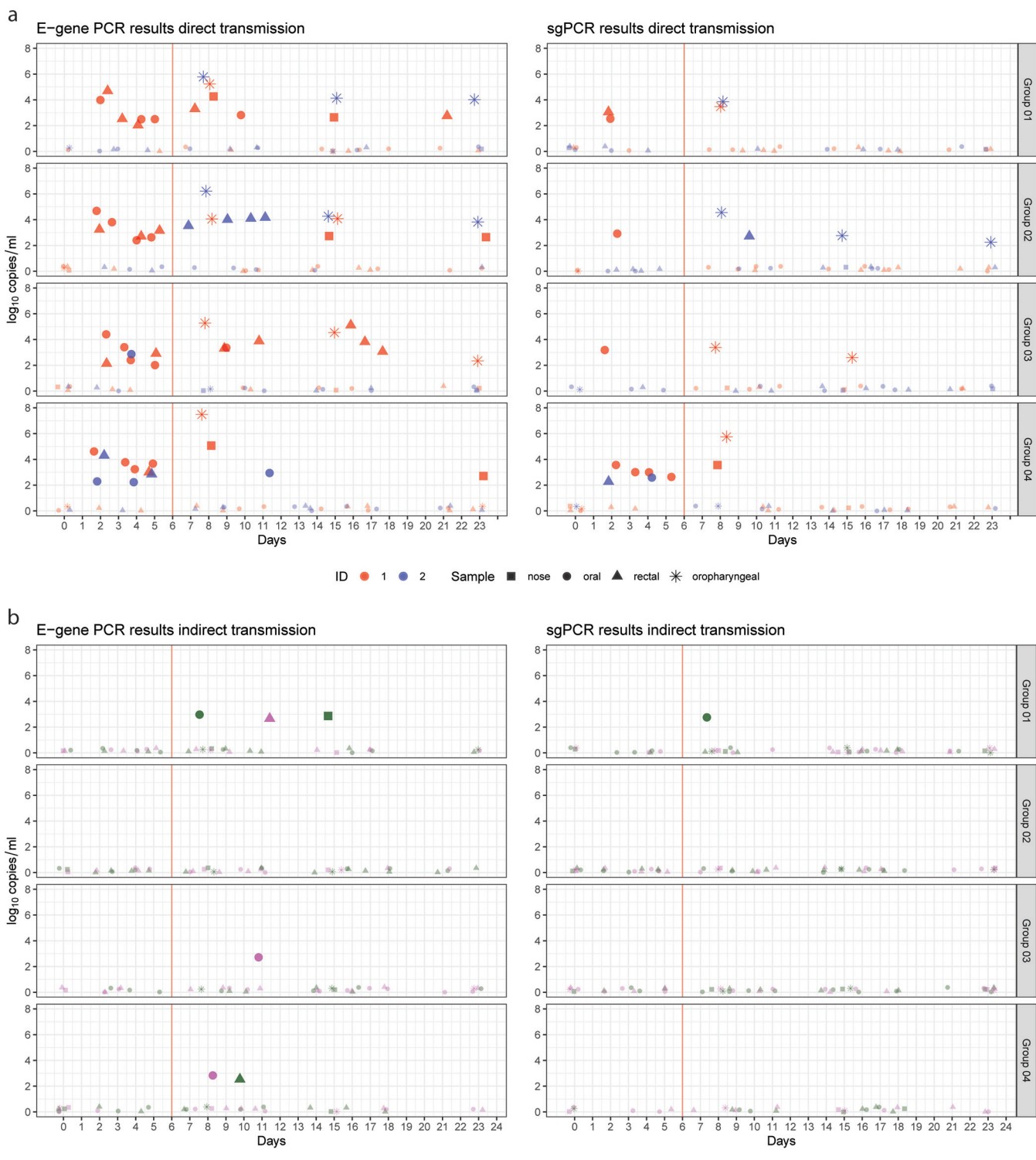

FIG 2 E-gene and subgenomic PCR results on swabs. For total E-gene PCR, the mean $\log_{10}$ RNA copies/mL from two technical replicates of the same swab are shown. The replicates were generated by isolating RNA from the swab sample in duplicate and subsequent PCR. The sgPCR was performed once with the RNA sample from the replicate that showed a lower Ct value in the total E-gene PCR. Shapes indicating the results are jittered so that overlapping shapes can still be observed. The red line indicates the day of removal of cats 1 and 2 from pen A and housing them in pen B. (a) Swabs from the direct transmission pairs. (b) Samples on the left of the red vertical line are collected from cats 3 and 4 (indirect transmission cats) before environmental exposure housed in pen B until D6, while samples on the right of the red vertical line are collected from cats 3 and 4 housed in contaminated pen A from D6 onwards.

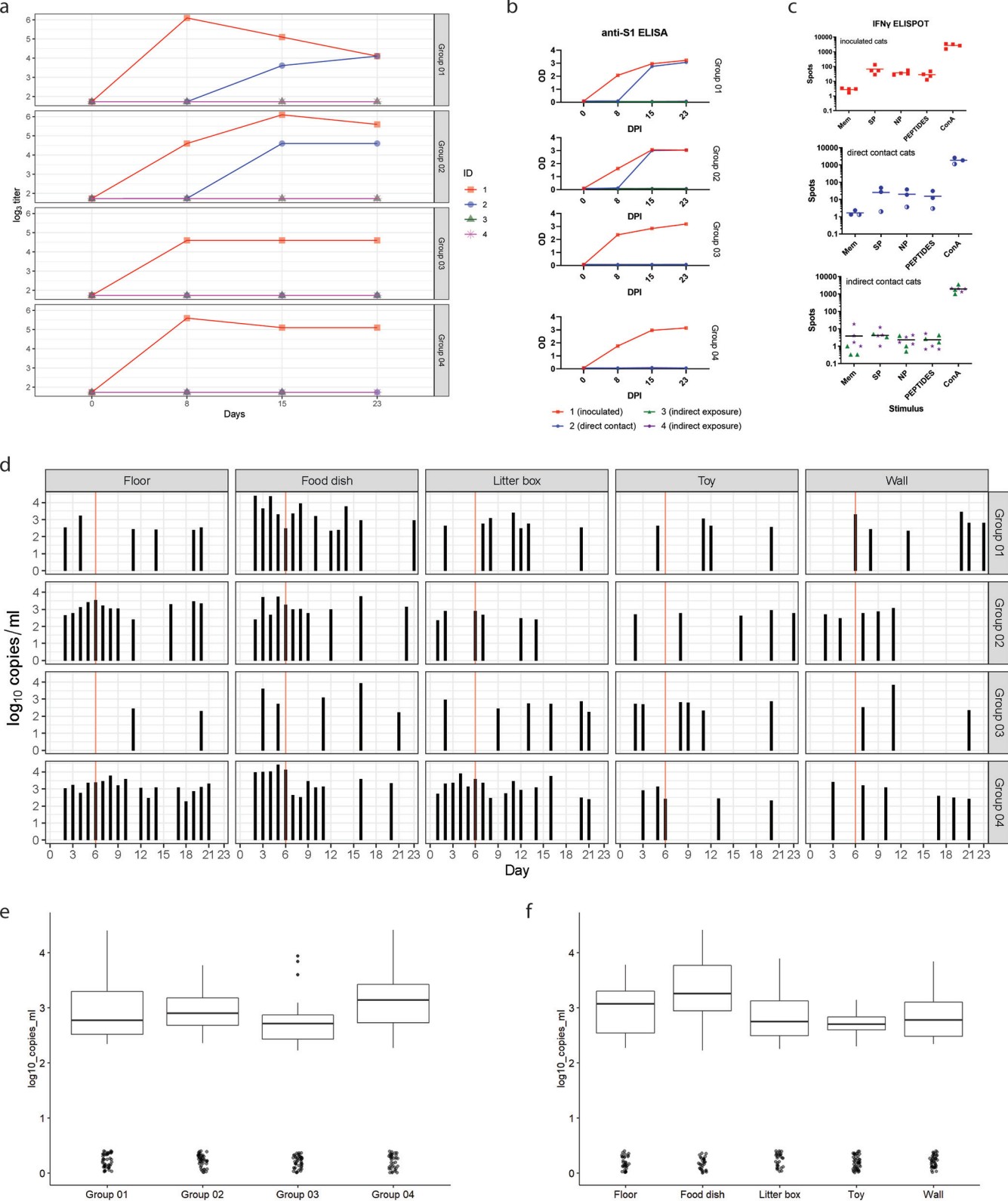

**FIG 3** Serology, immunology, and environmental contamination. (a) Neutralizing antibody titers expressed as 50% neutralization titer (MN$_{50}$). All four inoculated cats (cats X.1) and two of the direct contact cats (1.2 and 2.2) developed neutralizing antibodies, while the indirect exposure cats (cats X.3 and X.4) did not seroconvert. (b) Anti-S1 antibody response measured by ELISA at OD$_{420}$. Antibodies directed against S1 were detectable in all inoculated cats, as well as in the direct contact cats of group 1 and 2. (c) Interferon-γ response of PBMCs, collected on D23 upon necropsy, after stimulation for 48 h with spike protein (SP), nucleoprotein (NP), a peptide pool (PEPTIDES), and Concanavalin A (ConA) and medium (Mem) as control, measured by ELISpot analysis.

**TABLE 1** Results for the stochastic SIR model with environmental transmission[a]

| Scenario | Test | Data period | $\beta$ (day$^{-1}$) | $\mu$ (day$^{-1}$) | $R_{same\ day}$ | $R^b$ |
|---|---|---|---|---|---|---|
| Data Set 1 | sero | direct | 0.044 (0.013 to 0.105) | 0.001 | 0.264 (0.078 to 0.63) | 528.2 (156.1 to 1,260.4) |
| Data Set 2 | sero | both | 0.21 (0.06 to 0.49) | $\infty$ | 1.26 (0.37 to 3.02) | 1.26 (0.37 to 3.02) |
| Data Set 3 | SG | direct | 0.41 (0.16 to 0.85) | $\infty$ | 2.50 (0.97 to 5.15) | 2.50 (0.97 to 5.15) |
| Data Set 4 | SG | both | 0.23 (0.06 to 0.54) | 2.73 (0.77 to 15.82) | 1.38 (0.36 to 3.24) | 2.12 (0.92 to 4.08) |

[a]Four different scenarios (Data Sets) were analyzed based on the observed infection events using either seroconversion ("sero") or sgPCR ("SG") as a test to determine which cats had become infected and analyzing only the first part of the experiment with direct exposure ("direct") or combining the data with indirect exposure ("both"). The model yielded two parameters: the transmission rate parameter $\beta$ and the decay rate parameter $\mu$. From those parameters, the shedding rate $\varphi$ and the reproduction ratio $R$ of SARS-CoV-2 in cats were calculated, assuming an infectious period $T$ of 6 days (10).

[b]$R = \frac{\varphi}{\mu} \beta * T$ when both direct contact and the environment contribute to transmission or as $R = \beta * T$ when only "direct contact" transmission is considered, i.e., infection caused by virus shed on the same day.

of all sample types, the feeding dishes were those samples with the highest median genome copy numbers (Fig. 3e and f). The average temperature and relative humidity (RH) in the experimental pens over the course of the experiment were 19.65°C and 54.85%, respectively.

**Statistical analysis of transmission.** For transmission to be considered successful we defined contact cats as being infected and infectious either based on the sgPCR or on seroconversion. For quantification of the transmission rate and the decay rate parameters, the moment the donor and infected contact cats were identified as infected (moment of infection) and infectious was based on the E-gene PCR (evidence of exposure) because of the higher sensitivity of this PCR, i.e., the moment of infection was considered to be 1 to 2 days before the first E-gene-positive sample (latent period). For the analysis, they were included as possible source of infection from the first day they were observed as E-gene-PCR positive to the time their contact animal became infected (transmission took place). When transmission did not take place, a donor cat was considered infectious for the duration of observed E-gene-positive samples. However, in practice in these experiments the longest infectious period in the analysis used, based on these rules, was 7 days.

Table 1 details the maximum likelihood estimates of the parameters describing the transmission process for four different analyses using an extension of the direct transmission SIR model with environmental exposure. The analysis is performed on four different scenarios which are represented as Data Sets. Data Sets for scenarios "Data Set 1" and "Data Set 2" are based on using seroconversion as primary indicator for being infected and infectious, while "Data Set 3" and "Data Set 4" make use of the sgPCR results as primary indicator. "Data Set 1" and "Data set 3" only use data from the direct transmission pairs, while "Data set 2" and "Data set 4" also include the data from the environmental exposure pairs used to assess indirect transmission via the environment (see Supplementary Material 2 for all four Data Sets). The results of the analysis using Data Set 1 illustrates why the environmental SIR model is used. In this experiment, the direct contacts become infected very late in the experiment and two cats also escape infection, indicating that a build-up of infectiousness was required for infection to occur. This does not fit the direct transmission SIR model, which assumes that direct transmission infection would occur with equal probability on each day of the exposure. Hence, the environmental SIR model fits the data only with a low decay rate parameter. This estimate implies that after the direct exposure period ends, there would still be high infectivity, and cats added to the contaminated environment would get infected

**FIG 3** Legend (Continued)

An interferon-$\gamma$ response was observed in all inoculated cats and direct contact cats in group 1 and 2 after stimulation with SP, NP, the peptide pool, and ConA. No interferon-$\gamma$ response was detected in the direct contact cat of group 4 (indicated by half-filled bullets) or in the indirect contact cats, except for the positive control ConA. Cats 3.2 and 4.3 were not tested due to an insufficient number of PBMCs. (d to f) Environmental samples analyzed by E-gene PCR. (d) Viral RNA levels of positive environmental samples per group and sample type. Viral RNA was detectable throughout the study period. (e) Distribution of viral RNA load in the different groups. This graph shows the distribution of the viral E-gene RNA copy numbers (log$_{10}$) of samples collected in the four different experimental groups (group 1, 2, 3, or 4), independent of study day and independent of type of sample. (f) Distribution of viral RNA load per sample type. This graph shows the distribution of viral E-gene RNA copy numbers (log$_{10}$) of samples collected from the different sample locations (floor, food dish, litter box, toy or wall), independent of study day and independent of experimental group.

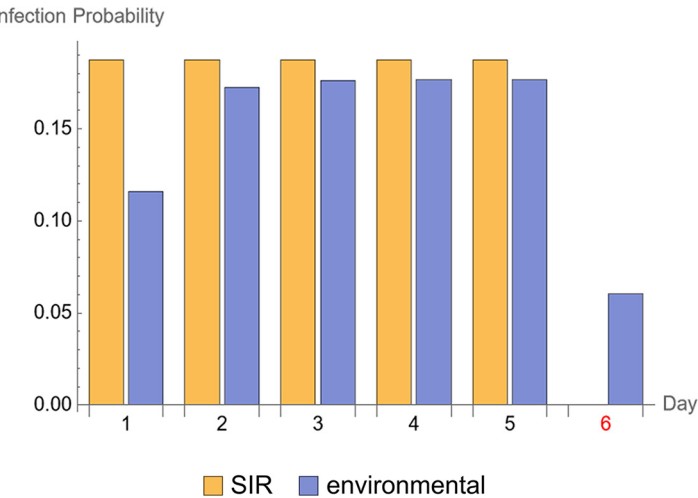

**FIG 4** Infection probability. The infection probability for each observed interval in the experiment where the day of the start of the interval is given on the X-axis. The results for two fitted stochastic models are given: "SIR" is the direct transmission SIR model and "environmental" is the environmental transmission SIR model. Day 6 is highlighted in red, because this is the day when the inoculated cat was removed from pen A.

readily. However, when we add the indirect exposure period to the data we see that none of the eight indirect exposed cats seroconverted. This is despite the presence of E-gene-PCR-positive material in the environment and in samples from the exposed cats for the whole experimental period (2 extra weeks) (Fig. 2 and 3). Using this Data Set (Data Set 2) the best fitting model estimates the decay rate parameter as very large, making the environmental SIR equivalent to the direct transmission SIR model.

For the analysis performed using the scenario represented in the data set "Data Set 3" we observed that the direct transmission SIR model fits the data best because the direct contact cat, cat 4.2 (Fig. 2), became infected very early during exposure at the interval 1 to 2 days. However, when using the sgPCR as indicator of infection, there is an infection observed (cat 1.3) in the indirect transmission group ("Data Set 4"). Using this scenario (Data Set 4), we get a finite estimate of the decay rate parameter. The maximum likelihood estimate (Table 1) is calculated for the transmission rate parameter $\beta = 0.23$ day$^{-1}$ (95% confidence intervals CI = 0.06 to 0.54) and for decay rate parameter $\mu = 2.73$ day$^{-1}$ (0.77 to 15.82), which implies an average survival time (duration of infectiousness in the environment = $1/\mu$) of 8.8 (1.5 to 31.2) hours. The interpretation of these estimates can be seen in Fig. 4. For the direct transmission SIR model, infection only occurs during direct exposure, i.e., in the 5 days the direct contact animals are present. In the environmental SIR model the infection probability increases over time during the direct exposure period and it is still present when the infected animals, contaminating the environment, are removed and naive animals are placed to indirectly expose them to the contaminated environment.

Assuming an infectious period of 6 days (10), the estimated $R_0$ is 2.12 (0.92 to 4.08) for transmission between cats for the direct and the indirect transmission route combined. The $R_0$ for the direct contact transmission alone is 1.38 (0.36 to 3.24). The contribution of the environment to the overall transmission is the difference between these two $R_0$: 2.12 − 1.38 = 0.74. In other words, around one-third of the overall transmission risk could be attributed to the environment.

## DISCUSSION

This study explored direct and indirect transmission of SARS-CoV-2, lineage B.1.22, among domestic cats in an experimental setting. We particularly quantified the duration of infectiousness of an environment with contaminated surfaces. We found that the infectiousness of contaminated surfaces would decay within 8.8 (95% CI = 1.5 to

31.2) hours, making transmission via contaminated surfaces alone inefficient, but it cannot be excluded yet. We also provide further confirmation that cat-to-cat transmission is efficient.

All four inoculated cats became infected with SARS-CoV-2 when measured by seroconversion or by sgPCR, and they remained clinically asymptomatic, except for mild nasal discharge in two cats. Compared to other studies focusing on transmission between cats, we used a lower inoculation dose (0.5 to 1.5 log lower than Gaudreault et al.; Shi et al.; Bao et al.; Bosco-Lauth et al. [7–9, 18]), and only the intranasal inoculation route. We did this because we hypothesized that a lower dose and intranasal exposure only would be a better approximation to natural infection of the donor cats. This hypothesis is based on published peak viral shedding titers of cats infected by direct contact, which is around $10^{4.0}$ PFU/mL (10). The absence of obvious clinical signs, with virus shedding measured in oral, nasal, oropharyngeal and rectal swabs, mild nonspecific histopathological findings, seroconversion of inoculated cats, as well as the transmission to naive cats by direct contact agrees with those other reports. The four inoculated cats transmitted SARS-CoV-2 to two (seroconversion) or three (sgPCR) direct contact cats, and all four direct contact cats had E-gene positive PCR samples.

A major challenge in the analysis of transmission experiments is the definition of "being infected." Almost all cats (except indirect contacts in group 3) were tested positive by E-gene PCR, indicating the presence of viral RNA in samples. These positive samples were further tested by sgPCR, which specifically quantifies the mRNA of the E-gene, generated during active virus replication. Next to the presence of viral RNA or mRNA, the third potential definition of "being infected" is seroconversion, which may be dependent on inoculation dose and follow-up time.

Given the different results in the number of contact cats considered infected based on serology or sgPCR, we analyzed transmission using either sgPCR or seroconversion as the determinant of successful infection. We explored the use of information generated from only the direct contact experiments or both direct and indirect contact experiments. Serology as an indicator for successful transmission did not provide sufficient information for reliable quantification of the decay rate. Contrasting $R_0$ estimates were obtained when using data from direct contact only (Data Set 1) and from direct and indirect contact (Data Set 2). However, when using sgPCR as an indicator for successful transmission, similar $R_0$ were estimated for Data Sets 3 and 4, which were 2.5 (95% CI = 0.97 to 5.15) and 2.12 (95% CI = 0.92 to 4.08), respectively (Table 1). These estimates are in agreement with those made by (10) using published data from direct transmission experiments $R_0 = 3.0$ (95% CI = 1.5 to 5.8) or from household infections $R_0 = 2.3$ (95% CI = 1.1 to 4.9) and provide further certainty that cat-to-cat transmission is efficient. In addition, the analysis also indicates that transmission is a result of both close contact interaction between cats and the contamination of the shared environment, with the later having a lower contribution (one-third of the overall contribution) to the risk of transmission.

Considering that not all direct contact cats seroconverted even though they tested positive by sgPCR, and hence are considered truly infected, our data suggest that surveys focusing on only seroprevalence in cats may be an underestimate of the true prevalence of SARS-CoV-2 in cats. Moreover, discrepancies between samples testing positive on one and negative on another assay can also be explained by varying sensitivities of the assays (14, 15).

Although our study contributes to an improved understanding of SARS-CoV-2 transmission among cats by providing a detailed estimation of both the transmission rate parameter and the decay rate parameter, there are limitations that need to be considered. For instance, we did not follow a potential transmission chain from an inoculated cat to a direct contact then to another direct contact cat, which would be a sensitive measure for infectiousness of the infected direct contact cats. Next, our follow-up time was limited to 17 days for the environmentally exposed cats, which may have been too short for cats exposed to lower amounts of SARS-CoV-2 to develop antibodies

compared to an inoculated cat. Furthermore, a sparse sampling under anesthesia was performed on purpose, however, samples collected from the nose and oropharynx were the most sensitive samples and we may have missed viral shedding of cats. This is particularly true for samples taken shortly after exposure, which usually have the highest viral loads and highest chance to isolate infectious virus. In our study, no samples suitable for virus isolation were obtained. Anesthesia can negatively impact the immune system (19, 20), implying that frequently anesthetized animals may become more sensitive to infections compared to animals infected under natural conditions. Finally, in our analysis we assume that direct contact cats are equally infectious as the inoculated cats. This assumption cannot be tested based on virus excretion because the samples that we took more frequently without anesthesia (oral swabs and rectal swabs) were less sensitive than samples taken sparsely under anesthesia (nasal swabs and oropharyngeal swabs), which would have been more adequate to compare the shedding levels of inoculated and direct contact cats. It also cannot be tested based on transmission to contact cats as only one possibly infected cat (the indirect contact cat) was housed together with a recipient. This did not lead to infection of the recipient cat, but there was also no infection in one out of four instances of direct contact between inoculated cats and their recipients.

Our analysis allowed the estimation of the expected duration of the infectiousness of a contaminated environment, with the probability of indirect transmission decaying within the first 8.8 h following contamination on average. This observation contrasts with the levels of RNA found in the assessed contaminated surfaces. Similar levels were observed during the whole experimental period (no apparent decay observed), which confirms that the presence of RNA does not reflect the risk of infection by exposure to contamination. These observations agree with laboratory experiments that showed very low to no environmental decay of RNA for up to 21 days on different surfaces (21), while the live virus could only be isolated from hours to few days depending on the surface and environmental conditions. In other words, estimated half-lives at room temperature and approximately 40% RH in contaminated stainless steel (same surface as the food dishes) ranged from 6 h to 43 h (22–24). While these virus survival studies suggest a potential risk of indirect transmission due to the presence of live virus on contaminated surfaces, our analysis directly assessed the probability of transmission and the duration of this risk. It should be noted that the estimated transmission parameters and duration of infectiousness of a contaminated environment may differ for other variants of SARS-CoV-2, for example, the omicron variant has been shown to survive longer than other SARS-CoV-2 variants (25), and like in people, infected animals may shed higher level of virus, and the transmissibility of this variant may be higher than the virus variant used for the experiment described here (26).

The estimated decay rate parameter for infectivity is an important finding as it is in line with *in vitro* measurements but not with the decay rate parameter for E-gene positivity in the environment. This finding may also help to further study cleaning and biosecurity measures in humans (e.g., 27).

In conclusion, our study provides further evidence for efficient transmission of SARS-CoV-2 between cats by direct contact even when a lower inoculation dose was chosen. In contrast, indirect transmission of SARS-CoV-2 to other cats via contaminated environment is less efficient, although it cannot be excluded.

## MATERIALS AND METHODS

**Ethical statement.** The study (experiment number 2020.D-0007.011) was performed under legislation of the Dutch Central Authority for Scientific procedures on Animals (CCD license no. AVD4010020209446), and approved by the Animal Welfare Body of Wageningen University and Research prior to the start of the in-life phase. Animal and laboratory work was performed in human biosafety level 3 laboratories at Wageningen Bioveterinary Research, Lelystad, the Netherlands.

**Virus.** SARS-CoV-2/human/NL/Lelystad/2020 was isolated from a human subject early in the pandemic and used for challenge inoculation. Details can be found in Gerhards et al. (16) and the sequence is available under GenBank accession number MZ144583.

**Animal housing and experimental design.** Eight male and eight female cats were obtained from Marshall Bioresources. They were 4 months of age, weighed between 1.8 and 2.8 kg upon arrival, were raised as SPF cats and were vaccinated against rabies and feline herpesvirus, feline calicivirus and feline panleukopenia virus. The acclimatization time was 11 days and the acclimatization time was also used to further socialize the animals to the animal technical staff. No blinding of personnel took place, and only healthy animals were included in the study.

Using the randomizer function in Microsoft Excel, cats were first divided by gender and then attributed to one of four experimental units with four cats per group (group 1, males; group 2, females; group 3, males; and group 4, females). Each group was housed in a separate hBSL3 animal unit, which in turn was divided in two pens A and B. Each pen had a volume of approximately 13.5m$^3$ (1.9 m width, 2.75 m length, and 2.58m height). Of each group (X = 1, 2, 3, or 4), cats X.1 and X.2 were housed in pen A, and the other two cats in pen B (cats X.3 and X.4) until study day D6. On D0, cat X.2 was temporarily removed from pen A for 1 day, and placed back on D1. On D6, cats X.1 and X.2 were removed from pen A and placed in pen B, while cats X.3 and X.4 were removed from pen B and placed in pen A until the end of the study (D23). In every animal unit, pen A and B were separated by more than 30 cm distance and a plastic separator to prevent droplet transmission between the pens. To avoid transmission by personnel, animal technicians first entered pen B before approaching pen A until D6, and from D7 onwards pen A first, and pen B second. The inoculated cats (i.e., X.1) were handled last. No exchange of equipment took place between pens. The pens were not cleaned from D0 until the end of the study, except for the daily removal of feces and urine from litter boxes. The average temperature throughout the study was 20.4°C in group 1 (range 19.9 to 21.1), 19.9°C in group 2 (range 18.3 to 21.0), 19.6°C in group 3 (range 19.0 to 20.3), and 18.7°C in group 4 (range 17.9 to 19.4). Relative humidity throughout the study was, on average, 51.3% in group 1 (range 43.0 to 68.0), 52.6% in group 2 (range 39.0 to 76.0), 52.5% in group 3 (range 42.0 to 71.0), and 63.0% in group 4 (range 46.0 to 77.0). There were 12 h of light and 12 h of darkness per day. Cats were provided with water and commercial cat pellets *ad libitum*. Feed was removed from the pens on the evenings before the days cats underwent anesthesia. Several types of commercial cat toys were available as enrichment, as well as hammocks, baskets, elevated resting spots, pillows, towels, and scratching posts.

**Animal inoculation and samplings.** All cats were anesthetized on D0, D8, and D15 by intramuscular injection of 0.04 mg/kg medetomidine and 4 mg/kg ketamine, which was antagonized by 0.1 mg/kg atipamezole intramuscularly. Anesthetics were injected in the hind legs, and the procedures were performed in the animal units.

On D0 in the morning, cats 1.1, 2.1, 3.1, and 4.1 were inoculated intranasally with $10^{4.5}$ TCID$_{50}$ SARS-CoV-2 under general anesthesia in a volume of 0.5 mL (0.25 mL per nostril, pipetted dropwise synchronous with the cat's breathing rhythm). Every cat was inoculated by a different animal technician. Cats were observed daily for their general health and respiratory signs post challenge. Body weight was measured regularly (D−11, D−10, D−7, D−5, D−3, D0, D2, D4, D7, D9, D11, D14, D16, D18, D21, and D23).

Nasal swabs and oropharyngeal swabs, as well as 2 mL serum and 1 mL heparinized blood were taken under general anesthesia on D0, D8, D15, and upon euthanasia. Oral and rectal swabs were taken without anesthesia on D0, D2 to D5, D7, D9 to D11, D14, D16 to D18, D21, and upon euthanasia.

Electrostatic dust cloths were used to sample the environment daily from D0 to D23. There were five different sample locations: floor (random area of 10 cm × 10 cm; a different spot was sampled every day), wall (random area of 1 m × 1 m; a different spot was sampled every day), litter box (nonporous plastic, rim of one litter box was wiped; the same rim of the same litter box was sampled every day), feeding tray (stainless steel, one feeding tray out of two was wiped before feed was added every day), toy (cotton and nonporous plastic; another randomly selected toy was wiped every day).

**Pathology and immunohistochemistry.** Cat 4.3 died on D18 unrelated to SARS-CoV-2 infection and was necropsied the same day. On D23, the remaining 15 cats were euthanized to perform a full macroscopic examination of all major organs and to collect blood and respiratory tract samples. Euthanasia was performed by intramuscular injection of anesthetics (0.06 mg/kg medetomidine and 6 mg/kg ketamine) in the hind leg, followed by blood sampling through the aorta and exsanguination through the brachiocephalic vein. The lungs were weighed after exsanguination, and expressed as percentage of the body weight on the day of necropsy. Tissue samples of the lungs (instilled with formaldehyde), nasal conchae, trachea, tracheobronchial lymph node, intestine (duodenum, ileum, and colon), mesenteric lymph node, and pancreas were taken in 4% neutral buffered formaldehyde for histopathology/immunohistochemistry (IHC) and frozen for virological analyses (lungs and conchae).

Formalin-fixed samples were embedded in paraffin, sectioned at 5 $\mu$m and stained with hematoxylin and eosin (H&E) for histological examination according to general pathology principles. IHC of histological specimens was performed with an antibody directed against the SARS-CoV nucleoprotein, as previously described (16).

**Organ suspensions, environmental samples, and swabs.** Lung and conchae samples were stored at ≤ −70°C until further processing. Each sample was weighed before homogenization in 6 mL of MEM, supplemented with 1% antibiotic/antimycotic solution (both from Gibco; Thermo Fisher Scientific; Waltham, MA, USA) in an Ultra Turrax Tube Drive (IKA; Staufen, Germany) for 30 s (lungs) or 50 s (conchae) at 6,000 rpm. Subsequently, homogenates were cleared by centrifugation at 3,400 × *g* at 4°C for 15 min. Cleared suspensions were mixed 1:3 with TRIzol-LS (Sigma; Saint Louis, MO, USA) and stored at ≤ −15°C until RNA isolation.

Environmental dust cloths were collected in individual seal bags and frozen directly at ≤ −70°C. To recover viral RNA, 10 mL of tissue culture medium (MEM) was added to the tissues and squidged vigorously, before taking a sample of 85 $\mu$L in 255 $\mu$L TRIzol-LS for RNA isolation.

Following sampling, all swabs were directly submerged in approximately 2 mL of tissue culture medium (MEM), kept on melting ice before vigorously vortexing for 30 s on a vortex (Labdancer, VWR International B.V., Amsterdam, the Netherlands), followed by centrifugation for 5 min at 1500 × g and 4°C. One sample aliquot of 200 µL was mixed with 200 µL lysis buffer (Molgen, Veenendaal, the Netherlands), and stored at ≤ −15°C until RNA isolation.

**RNA extraction and PCR.** RNA of samples stored in TRIzol-LS was extracted using Direct-zol RNA MiniPrep kit (Zymo Research, RefNo R1013; Irvine, CA, USA) according to the manufacturer's instructions, without DNase treatment. RNA of samples stored in Molgen lysis buffer was isolated by an automated robot system (PurePrep 96) using the Molgen RNA isolation kit (OE00290096). Isolated RNA was stored at ≤ −70°C.

Viral RNA was detected with a primer and probe set annealing to the E gene according to Corman et al. (14). Subgenomic viral RNA was detected with a prime and probe set as described in Wolfel et al. (15). PCR results were expressed as $Log_{10}$ viral RNA copies, based on standard curves, as described previously (16).

Because of the high Ct values, all swabs were isolated and tested by E-gen PCR two times. Swab samples that yielded a Ct value only once were isolated and tested a third time. Only those swabs that tested positive twice were reported as positive samples and were subsequently tested by subgenomic PCR.

**Detection of neutralizing antibodies.** Collected blood samples were separated by centrifugation for 10 min at 1,250 × g at room temperature (RT) after clotting for at least 1 h at RT. Resultant sera were stored at ≤ −15°C before heat inactivation for 2 h at 56°C.

Wild-type virus neutralization tests (VNT) and immuno-Peroxidase Monolayer Assay (IPMA) were performed on Vero E6 cells (ATCC CRL-1586; Manassas, VA, USA) in technical duplicates by 3-fold serial dilutions on 96-well plates, as described previously (16). Briefly, serum was initially diluted 1:10 and 50 µL of each sample were added to 50 µL SARS-CoV-2 (~100 $TCID_{50}$) in MEM. After an incubation step of 1.5 h, 15,000 Vero-E6 cells were added to each well. After 4 days, plates were fixed with 4% formaldehyde and permeabilized with ice-cold 100% methanol.

To visualize SARS-CoV-2, plates were additionally permeabilized with 1% Triton X-100 and unspecific binding was blocked with 5% normal horse serum in PBS. A rabbit antiserum (rabbit-anti-SARS-CoV-2-S1-2ST [619F], Davids Biotechnologie GmbH) was used to detect the viral spike protein. After incubation with goat-anti-rabbit-HRP (Dako; Agilent; Santa Clara, CA, USA), followed by addition of AEC (3-Amino-9-ethylcarbazole) substrate solution, a clear red-brown color developed within 30 to 40 min, which was subsequently evaluated under a standard light microscope.

The titer of each duplicate was calculated as the average of the reciprocal value of the last dilution that showed at least 50% neutralization, after log transformation. The titers are expressed as virus microneutralization titer 50 ($MN_{50}$).

**ELISA.** Anti-S1 ELISA of sera was performed as described previously (28). Briefly, sera were diluted 1:50, added to S1-protein coated microtiter plates and incubated for 1 h at 37°C before incubation with horseradish-peroxidase conjugated secondary goat-anti-cat IgG/HRP (dilution 1:4,000, Rockland Immunochemicals Inc., Limerick, PA, USA) for 1 h at 37°C. The reaction was visualized using 3,3′,5,5′-tetramethylbenzidine, quenched by sulfuric acid and optical density measurement at 450 nm.

**PBMC isolation and interferon-γ-ELISpot.** To isolate peripheral blood mononuclear cells (PBMCs), 15 mL ficoll (GE Healthcare Bio Sciences B.V. Sweden) was added to leucosep tubes (GREINER BIO-ONE B.V., Germany), and spun down for 1 min at 1,000 × g. A total of 15 mL of heparinized blood was diluted 1:1 with RPMI 1640 (Life Technologies, the Netherlands) and added to the leucosep tubes containing ficoll. After centrifugation for 10 min at 1,000 × g, the interface was harvested, diluted 1:1 with cell culture medium and spun down for 7 min at 400 × g. Next, cells were incubated with 4,500 µL ACK lysis buffer (Life Technologies, USA) for 5 min at room temperature and spun down again for 7 min at 400 × g. The resultant pellet was resuspended in 10 mL cell culture medium and cells were counted. Then, $2 × 10^5$ cells were added to an ELISpot plate, prepared according to manufacturer's instructions (Cat IFN-γ ELISpot Plus Kit, MABTech, Nacka Strand, Sweden) containing 100 µL culture medium with or without stimulus. The following stimuli were used: Spike protein 30 µg/mL (SARS-CoV-2 (2019-nCoV) Spike Protein (S1+S2 ECD, His tag), Sino Biological, Eschborn, Germany), nucleoprotein 10 µg/mL (SARS-CoV-2 (2019-nCoV) NucleocapsidProtein (His tag), Sino Biological, Eschborn, Germany), SARS-CoV-2 Spike protein peptide pool 6 µg/mL (Pepscan Presto, Lelystad, the Netherlands), concanavalin A 20 µg/mL (Merck, Germany). The cells were then incubated for 48 h at 37°C + 5%$CO_2$ and spots were developed according to manufacturer's instructions. Spots were analyzed by an ELISpot reader (VSpot Spectrum, AID GmbH, Germany).

**Statistical analyses incorporating environmental transmission.** The observed number of cases in an interval in each separated pen is the dependent variable. This dependent variable is assumed to be binomial distributed with the number of noninfected cats as the binomial total. Additionally, it is assumed that inoculated and contact infected cats are equally infectious. In this case, analysis can be done as Bernoulli trials for each recipient separately. Whether the recipient becomes infected also depends on the number of infected cats in the pen through the infectivity present in the environment. The environmental load is assumed to change deterministically according to the following differential equation:

$$\frac{d\,E(t)}{dt} = \varphi\,It - \mu\,E(t) \qquad (1)$$

Note that "$It$" is always a whole number and changes stochastically with integer jumps. Hence, the different notation used for $E(t)$ which changes continuously. Given the "$It$" and "$E0$" (the environmental

contamination at the start of the interval), values $E(t)$ can be calculated during each interval by solving equation 1. The value of $\varphi$ can be fixed without loss of generality.

To further understand the contribution of a contaminated environment to transmission, the shedding rate $\varphi$ is chosen such that $\int_{t}^{t+1} E(t)\ dt = 1$ for one infectious individual during 1 day in a clean environment. This shedding rate then depends on the decay rate $\mu$ as follow:

$$\varphi(\mu) = \frac{\mu^2}{-1 + e^{-\mu} + \mu} \tag{2}$$

For an interval $\Delta t$, the probability to become infected for each of the recipient cats is:

$$p(\mu, \beta) = 1 - e^{-\beta \frac{\int_{0}^{\Delta t} E(t)dt}{N}} \tag{3}$$

This model assumes shedding to the environment with constant rate parameter $\varphi$ and decay rate parameter (inactivation of the virus) $\mu$. The transmission occurs with transmission rate parameter $\beta$. The parameters $\mu$ and $\beta$ can be estimated by maximum likelihood either directly by optimizing the log likelihood as we did here (Supplementary Material 2) or by generalized linear models using an offset (29). The parameter $\varphi$ can be then calculated using the estimated value of $\mu$.

The total contribution of both environment and direct contact to transmission from the first day of shedding of one infectious individual is then $\frac{\varphi}{\mu} \beta$ and the basic reproduction number is:

$$R_0 = \frac{\varphi}{\mu} \beta * T \tag{4}$$

where the $T$ is the infectious period. Because for the first day of exposure (donor – recipient) $\int_{t}^{t+1} E(t)dt = 1$, the transmission rate during this day is equal to $\beta$ and it represents the expected transmission due to direct contact alone. The reproduction number $R$ due to direct contact alone would be $R = \beta * T$. The apparent contribution of the environment to the total transmission is then the difference between $R_0$ and $R$.

More details of the formulas used are given in Supplementary Material 2.

## SUPPLEMENTAL MATERIAL

Supplemental material is available online only.
**SUPPLEMENTAL FILE 1**, PDF file, 0.8 MB.

## ACKNOWLEDGMENTS

This study was commissioned by the Dutch Ministry of Agriculture and Food under the ARVODI-2018 agreement concerning policy supporting research on COVID-19 in cats and dogs.

We are thankful to Rineke de Jong, Stéphanie Vastenhouw, Sophie van Oort, José Harders-Westerveen, Bregtje Smid, Rianka Vloet, Romy Dresken, Julia Antonissen, Judith Bonsing, Ralph Kok, Pieter Roskam, Patrik Brandenburg, and Corry Dolstra for their excellent technical contribution. We thank the staff of WBVR's animal facility and biosafety officers. We also thank Berend Jan Bosch and Wentao Li (University of Utrecht, the Netherlands) for providing us the anti-spike S1 antibody used in this study for staining of the cell monolayers in the IPMA. Illustrations in Fig. 1 were created with BioRender.com.

We declare that there are no competing interests.

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
