## [Reviewer comments · Microbiology Spectrum]

Microbiology Spectrum

Efficient direct and limited environmental transmission of SARS-CoV-2 lineage B.1.22 in domestic cats

Nora Gerhards, Jose Gonzales, Sandra Vreman, Lars Ravesloot, Judith van den Brand, Harmen Doekes, Herman Egberink, Arjan Stegeman, Nadia Oreshkova, WH van der Poel, and Mart de Jong

Corresponding Author(s): Nora Gerhards, Wageningen Bioveterinary Research

Review Timeline:

Submission Date:	July 5, 2022
Editorial Decision:	September 6, 2022
Revision Received:	October 19, 2022
Editorial Decision:	February 7, 2023
Revision Received:	February 24, 2023
Accepted:	May 4, 2023

Editor: Rafael A. Medina

Reviewer(s): Disclosure of reviewer identity is with reference to reviewer comments included in decision letter(s). The following individuals involved in review of your submission have agreed to reveal their identity: Victor Neira-Ramirez (Reviewer #1)

Transaction Report:

DOI: <https://doi.org/10.1128/spectrum.02553-22>

September 6, 2022

Dr. Nora M. Gerhards
Wageningen Bioveterinary Research
Epidemiology, Bioinformatics & animal models
Houtribweg
39
Lelystad, Flevoland 8221 RA
Netherlands

Re: Spectrum02553-22 (Efficient direct and limited environmental transmission of SARS-CoV-2 lineage B.1.22 in domestic cats)

Dear Dr. Nora M. Gerhards:

You will see from the reviewers' comments that additional information needs to be provided for clarity. Please, specifically address the comments in regards to the qRT-PCR results and the interpretation of limited transmission. Also, please provide information on whether viral isolation was attempted. In materials and methods please include further details on the environmental sampling performed. We ask that this be provided, before we consider you manuscript for publication.

Link Not Available

Sincerely,

Rafael A. Medina

Journals Department
Reviewer comments:

Reviewer #1 (Comments for the Author):

In this study, the authors studied the direct and indirect transmission of SARS-CoV-2 by fomites. The transmission by direct contact was efficient, while indirect transmission by a contaminated environment was substantively less efficient. In general, the study has been adequately designed and the materials and methods sustained the results and discussion.

General comments:

- * Was it possible to perform virus isolation (VI)?
- * It will be valuable to include more details on the M&M of environmental sampling, since it is the main focus of the study. I.e. In my understanding, the authors wiped a tissue over the surfaces to collect the sample. Were the samples collect in the same spot over the duration of the experiment?
- * Were the cages cleaned daily? Sandboxes?
- * Describe the type of surfaces used (e.g. porous and non-porous). Discuss the virus survival depending on surfaces contaminated in the study.
- * About the lack of direct transmission in one group. Do you have records of cat behavior? Did the cats lick each other?
- * Did the authors change the sandboxes?
- * Include a paragraph on the limitations of this study, especially the absence of VI.
- * Review the manuscript for proper use of English.

Reviewer #2 (Comments for the Author):

Gerhards et al. have performed experimental studies to evaluate both direct contact and environmental transmission via fomites of SARS-CoV-2 in cats. They found direct contact transmission occurred to 3/4 contacts with limited evidence of environmental transmission to a single cat. The animals were sampled repeated and both qRT-PCR and seroconversion were used as evidence of infection. As a final component of the manuscript, the author's utilize their experimental data to estimate the R0, decay rate parameter.

Major Comments:

1. The authors have extensively used qRT-PCR to assess viral infection. They have used both E gene PCR which indicates the presence of the virus and sub genomic PCR as a surrogate for viral replication. When using qRT-PCR to assess viral replication the data is convincing when viral RNA is detected over multiple time points and the levels of RNA peak and then decline. This would be consistent with a productive infection if replicating virus was assayed. The interpretation of E and sgPCR becomes challenging when there is detection at a single time point. This is observed in the Group 1 indirect contact. This animal also did not seroconvert. Therefore, the conclusion of limited transmission is based upon a single data point. It is possible that an infected animal could have shed mucus containing infected cells and that the indirect contact animal sampled this infected material. Thus, I would recommend that all the indirect contact samples be cultured to ensure replicating virus can be recovered. In the event infectious virus cannot be recovered the authors will need to explain two possible interpretations of their findings: one in which the sample is positive and one in which the sample is virus negative.

Minor Comments:

1. Results section page 4. The experimental designed outlined in this section is very difficult to understand. Understanding this section required re-reading both the results, methods section, and figure legend multiple times. This section should be revised for clarity to clearly explain the experimental design and should also clearly indicate the number of animals and replicates.
2. In the section on clinical signs and pathology, the authors need to indicate if they expected pathology on day 23. If so, then please provide a rationale and/or reference. If not, please indicate pathology would not be expected at day 23.
3. In general, the figures in panel 2 are difficult to interpret. It would be easier if in panel B the figure were divided to show titers in the transmission pairs separately from the environmental transmission animals.
4. Figure 3e and f. Please clarify what the box plots and the data points indicate.
5. Discussion, last line of the first paragraph. Please revise. The authors do not demonstrate sustained transmission as they only evaluated a single transmission event in 4 pairs of ferrets. To demonstrate sustained transmission and chain of transmission is required.

Staff Comments:

Preparing Revision Guidelines

Please return the manuscript within 60 days; if you cannot complete the modification within this time period, please contact me. If you do not wish to modify the manuscript and prefer to submit it to another journal, please notify me of your decision immediately so that the manuscript may be formally withdrawn from consideration by Microbiology Spectrum.

Spectrum02553-22 (Efficient direct and limited environmental transmission of SARS-CoV-2 lineage B.1.22 in domestic cats)

Response to reviewers

Reviewer comments:

Reviewer #1 (Comments for the Author):

In this study, the authors studied the direct and indirect transmission of SARS-CoV-2 by fomites. The transmission by direct contact was efficient, while indirect transmission by a contaminated environment was substantively less efficient. In general, the study has been adequately designed and the materials and methods sustained the results and discussion.

General comments:

* Was it possible to perform virus isolation (VI)?

Response of authors: We thank the reviewer for this comment and we agree that virus isolations would have made our manuscript much stronger. However, we did not attempt to isolate virus from the swab samples. This has several reasons: (1) The ct values of positive swabs were >25 and frequently around 30, which is rather high for isolating infectious virus. (2) Our virus isolation assay may be less sensitive to VI assays of other labs, as we could not retrieve infectious virus from oropharyngeal swabs of hamsters either (Gerhards et al, 2021). (3) In another still unpublished study, we have euthanized cats within the first 14 days post infection. We could isolate infectious virus from lungs and trachea up until DPI 4, and from the nasal conchae until DPI 7. Considering that the first swabs were taken on DPI 8 in the current study, we think that our study designs has missed the window to take samples for virus isolations. This limitation of our study design is also described the discussion section.

We added a sentence in the results section clarifying this comment:

Viral RNA load in swabs and organs

Oral and rectal swabs were taken frequently, while nasal and oropharyngeal swabs were only sampled on a few time points (D0, D8, D15 and D23). Swabs were analyzed by both total E-gene PCR according to (14), and by subgenomic PCR (sgPCR) according to (15). **No virus isolations were attempted, because many swab samples had high Ct values (> 30) from which virus can only occasionally be isolated (16, 17).**

(...)

And we also added a sentence in the discussion section:

(...) Although our study contributes to an improved understanding of SARS-CoV-2 transmission amongst cats by providing a detailed estimation of both the transmission rate parameter and the decay rate parameter, **there are limitations that need to be considered.** For instance, we did not follow a potential transmission chain from an inoculated cat to a direct contact then to another direct contact cat, which would be a sensitive measure for infectiousness of the infected direct contact cats. Next, our follow-up time was limited to 17 days for the environmentally exposed cats, which may have been too short for cats exposed to lower amounts of SARS-CoV-2 to develop antibodies compared to an inoculated cat. Furthermore, a sparse sampling under anesthesia was performed on purpose, however, samples collected from the nose and oropharynx were the most sensitive samples and we may have missed viral shedding of cats. **This is particularly true for samples taken shortly after exposure, which usually have the highest viral loads and highest chance to retrieve an infectious virus. In our study, no samples suitable for virus isolation were obtained.** (...)

* It will be valuable to include more details on the M&M of environmental sampling, since it is the main focus of the study. I.e. In my understanding, the authors wiped a tissue over the surfaces to collect the sample. Were the samples collect in the same spot over the duration of the experiment?

Response of authors: The samples were all collected on different spots each day, except of the litter box. We specified this in the methods section:

Animal inoculation and samplings

(...) Electrostatic dust cloths were used to sample the environment daily from D0 to 23. There were five different sample locations: floor (random area of 10x10cm; a different spot was sampled every day), wall (random area of 1x1m; a different spot was sampled every day), litter box (non-porous plastic, rim of one litter box was wiped; the same rim of the same litter box was sampled every day), feeding tray (stainless steel, one feeding tray but of two was wiped before feed was added per group every day), toy (cotton and non-porous plastic; another randomly selected toy was wiped every day). (...)

* Were the cages cleaned daily? Sandboxes?

Response of authors: The pens were not cleaned after cats X.1 were inoculated to not interfere with environmental contamination. Faeces and urine were removed from the litter boxes each day.

We added this to the methods section:

Animal housing and experimental design

(...) In every animal unit, pen A and B were separated by more than 30cm distance and a plastic separator to prevent droplet transmission between the pens. To avoid transmission by personnel, animal technicians first entered Pen B before approaching Pen A until D6, and from D7 onwards Pen A first, and Pen B second. The inoculated cats (i.e. X.1) were handled last. No exchange of equipment took place between pens. The pens were not cleaned from D0 until the end of the study, except for the daily removal of feces and urine from litter boxes. The average temperature throughout the study was 20.4°C in Group 1 (range 19.9 - 21.1), 19.9°C in Group 2 (range 18.3 - 21.0), 19.6°C in Group 3 (range 19.0 – 20.3) and 18.7°C in Group 4 (range 17.9 – 19.4). (...)

* Describe the type of surfaces used (e.g. porous and non-porous). Discuss the virus survival depending on surfaces contaminated in the study.

Response of authors: In the materials and methods section, we added the texture of plastic. For stainless steel, floor and walls as well as for cotton, a distinction between porous and non-porous is not common.

Animal inoculation and samplings

(...) Electrostatic dust cloths were used to sample the environment daily from D0 to 23. There were five different sample locations: floor (random area of 10x10cm; a different spot was sampled every day), wall (random area of 1x1m; a different spot was sampled every day), litter box (non-porous plastic, rim of one litter box was wiped; the same rim of the same litter box was sampled every day), feeding tray (stainless steel, one feeding tray was wiped before feed was added; one out of the two feeding trays were sampled per group every day), toy (cotton and non-porous plastic; another randomly selected toy was wiped every day). (...)

* About the lack of direct transmission in one group. Do you have records of cat behavior? Did the cats lick each other?

Response of authors: As described in the supplemental material 1, we recorded cat behaviour using video, and analyzed the activity in each group based on pixel changes. Compared to the other three groups, it seems that overall activity was somewhat lower in group 3. However, due to the three-dimensional animal housing, it is difficult to distinguish between real interaction and

false interaction due to proximity of two cats on a two-dimensional video, when they in fact were on different levels on the z-axis. Moreover, the analysis of interactions in this study is further complicated by the fact that some interactions are not visible on video (e.g. when two cats are in the same litter box). For proper analysis of interactions, a tracker-based system would have been preferred that allows analyses in a three-dimensional environment. We therefore feel that we could only speculate about a reduced interaction in group 3, without sufficient proof supporting our speculation.

* Did the authors change the sandboxes?

Response of authors: Please see our response to the question regarding cleaning of the pens above.

* Include a paragraph on the limitations of this study, especially the absence of VI.

Response of authors: We added the absence of virus isolation to our section of limitations in the discussion:

(...)

Although our study contributes to an improved understanding of SARS-CoV-2 transmission amongst cats by providing a detailed estimation of both the transmission rate parameter and the decay rate parameter, **there are limitations that need to be considered.** For instance, we did not follow a potential transmission chain from an inoculated cat to a direct contact then to another direct contact cat, which would be a sensitive measure for infectiousness of the infected direct contact cats. Next, our follow-up time was limited to 17 days for the environmentally exposed cats, which may have been too short for cats exposed to lower amounts of SARS-CoV-2 to develop antibodies compared to an inoculated cat. Furthermore, a sparse sampling under anesthesia was performed on purpose, however, samples collected from the nose and oropharynx were the most sensitive samples and we may have missed viral shedding of cats. **This is particularly true for samples taken shortly after exposure, which usually have the highest viral loads and highest chance to retrieve an infectious virus. The absence of suitable samples for virus isolation is a clear limitation of our study.** Anesthesia can negatively impact the immune system (19, 20), implying that frequently anesthetized animals may become more sensitive to infections compared to animals infected under natural conditions. Finally, in our analysis we assume that direct contact cats are equally infectious as the inoculated cats. This assumption cannot be tested based on virus excretion because the samples that we took more frequently without anesthesia (oral swabs and rectal swabs) were less sensitive than samples taken sparsely under anesthesia (nasal swabs and oropharyngeal swabs), which would have been more adequate to compare the shedding levels of inoculated and direct contact cats. It also cannot be tested based on transmission to contact cats as only one possibly infected cat (the indirect contact cat) was housed together with a recipient. This did not lead to infection of the recipient cat, but there was also no infection in one out of four instances of contact between inoculated cats and their recipients.

(...)

* Review the manuscript for proper use of English.

Response of authors: The manuscript was edited by a language editing office to ensure proper use of English. The textual modifications are not highlighted in the revised version of the manuscript, to maintain readability.

Reviewer #2 (Comments for the Author):

Gerhards et al. have performed experimental studies to evaluate both direct contact and environmental transmission via fomites of SARS-CoV-2 in cats. They found direct contact transmission occurred to 3/4 contacts with limited evidence of environmental transmission to a single cat. The animals were sampled repeated and both qRT-PCR and seroconversion were used as evidence of infection. As a final component of the manuscript, the author's utilize their experimental data to estimate the R_0 , decay rate parameter.

Major Comments:

1. The authors have extensively used qRT-PCR to assess viral infection. They have used both E gene PCR which indicates the presence of the virus and sub genomic PCR as a surrogate for viral replication. When using qRT-PCR to assess viral replication the data is convincing when viral RNA is detected over multiple time points and the levels of RNA peak and then decline. This would be consistent with a productive infection if replicating virus was assayed. The interpretation of E and sgPCR becomes challenging when there is detection at a single time point. This is observed in the Group 1 indirect contact. This animal also did not seroconvert. Therefore, the conclusion of limited transmission is based upon a single data point. It is possible that an infected animal could have shed mucus containing infected cells and that the indirect contact animal sampled this infected material. Thus, I would recommend that all the indirect contact samples be cultured to ensure replicating virus can be recovered. In the event infectious virus cannot be recovered the authors will need to explain two possible interpretations of their findings: one in which the sample is positive and one in which the sample is virus negative.

Response of authors: We thank the reviewer for this comment and agree to the need for virus isolations. Please see our answer to the first question of reviewer 1 for explanation why virus isolations were not performed. Moreover, we have analyzed the data using the two different scenarios as presented in table 1 – if we use serology as a readout, the indirect contacts did not become infected as well as 2/4 direct contacts; while if we use sgPCR as a readout, 1/8 indirect contacts as well as 3/4 direct contacts are considered infected.

We added another column in table 1 'R same day' to indicate the transmission rate that occurs within one day which we hope will aid understanding the differences in the estimates:

Table 1. Results for the stochastic SIR model with environmental transmission. Four different scenarios (datasets) were analyzed based on the observed infection events using either seroconversion ('sero') or sgPCR ('SG') as a test to determine which cats had become infected and analyzing only the first part of the experiment with direct exposure ('direct') or combining the data with indirect exposure ('both'). The model yielded two parameters: the transmission rate parameter β and the decay rate parameter μ . From those parameters, the shedding rate φ and the reproduction ratio R of SARS-Cov-2 in cats were calculated, assuming an infectious period T of six days (10).

Scenario	Test	Data period	β (day ⁻¹)	μ (day ⁻¹)	$R_{\text{same day}}$	R^a
data1	sero	direct	0.044 (0.013-0.105)	0.001	0.264 (0.078-0.63)	528.2 (156.1-1260.4)
data2	sero	both	0.21 (0.06-0.49)	∞	1.26 (0.37-3.02)	1.26 (0.37-3.02)
data3	SG	direct	0.41 (0.16-0.85)	∞	2.50 (0.97-5.15)	2.50 (0.97-5.15)
data4	SG	both	0.23 (0.06-0.54)	2.73 (0.77-15.82)	1.38 (0.36-3.24)	2.12 (0.92-4.08)

$R = \frac{\phi}{\mu} \beta * T$ when both direct contact and the environment contribute to transmission or as $R = \beta * T$ when only "direct contact" transmission is considered, i.e. infection caused by virus shed on the same day.

The results section was changed as following:

(...)

For the analysis performed using the scenario represented in the datasets 'data3' we observed that the direct transmission SIR model fits the data best because the direct-contact cat - Cat 4.2 (Figure 2) - became infected very early during exposure at the interval 1 to 2 days. However, when using the sgPCR as indicator of infection, there is an infection observed (Cat 1.3) in the indirect transmission group ('data4'). Using this scenario (data4), we get a finite estimate of the decay rate parameter. The maximum likelihood estimate (Table 1) is calculated for the transmission rate parameter $\beta = 0.23 \text{ day}^{-1}$ (95% confidence Intervals CI: 0.06 - 0.54) and for decay rate parameter $\mu = 2.73 \text{ day}^{-1}$ (0.77 - 15.82), which implies an average survival time (duration of infectiousness in the environment = $1/\mu$) of 8.8 (1.5 - 31.2) hours. The interpretation of these estimates can be seen in Figure 4. For the direct transmission SIR model, infection only occurs during direct exposure, i.e. in the five days the direct contact animals are present. In the environmental SIR model the infection probability increases over time during the direct exposure period and it is still present when the infected animals, contaminating the environment, are removed and naïve animals are placed to indirectly expose them to the contaminated environment.

Assuming an infectious period of 6 days (10), the estimated R_0 is 2.12 (0.92 - 4.08) for transmission between cats for the direct and the indirect transmission route combined. The R_0 for the direct contact transmission alone is 1.38 (0.36 - 3.24). The contribution of the environment to the overall transmission is the difference between these two R_0 : $2.12 - 1.38 = 0.74$. In other words, around 1/3 of the overall transmission risk could be attributed to the environment. (...)

And in the discussion, we added:

(...) A major challenge in the analysis of transmission experiments is the definition of 'being infected'. Almost all cats (except indirect contacts in Group 3) were tested positive by E-gene PCR, indicating the presence of viral RNA in samples. These positive samples were further tested by sgPCR, which specifically quantifies the mRNA of the E-gene, generated during virus active replication. Next to the presence of viral RNA or mRNA, the third potential definition of 'being infected' is seroconversion, which may be dependent on inoculation dose and follow-up time.

Given the different results in the number of contact cats considered infected based on serology or sgPCR, we analysed transmission using either sgPCR or seroconversion as the determinant of successful infection. We explored the use of information generated from only the direct-contact experiments or both direct and indirect contact experiments. Serology as an indicator for successful transmission did not provide sufficient information for reliable quantification of the decay rate. Contrasting R_0 estimates were obtained when using data from direct-contact only (data1) and from direct and indirect contact (data2). However, when using sgPCR as an indicator for successful transmission, similar R_0 were estimated for datasets 3 and 4, which were 2.5 (95%CI: 0.97 - 5.15) and 2.12 (95%CI: 0.92 - 4.08), respectively (Table 1). These estimates are in agreement with those made by (10) using published data from direct transmission experiments $R_0 = 3.0$ (95% CI: 1.5 - 5.8) or from household infections $R_0 = 2.3$ (95%CI: 1.1 - 4.9) and provide further certainty that cat-to-cat transmission is efficient. In addition, the analysis also indicates that transmission is a result of both close contact interaction between cats and the contamination of the shared environment, with the later having a lower contribution (1/3 of the overall contribution) to the risk of transmission. (...)

And in the methods section, we added:

Statistical analyses incorporating environmental transmission

The observed number of cases in an interval in each separated pen is the dependent variable. This dependent variable is assumed to be binomial distributed with the number of non-infected cats as the binomial total. Additionally, it is assumed that inoculated and contact infected cats are equally infectious. In this case, analysis can be done as Bernoulli trials for each recipient separately. Whether the recipient becomes infected also depends on the number of infected cats in the pen through the infectivity present in the environment. The environmental load is assumed to change deterministically according to the following differential equation:

$$\frac{dE(t)}{dt} = \varphi It - \mu E(t) \quad (1)$$

Note that 'It' is always a whole number and changes stochastically with integer jumps. Hence the different notation used for E(t) which changes continuously. Given the 'It' and 'E0' (the environmental contamination at the start of the interval), values E(t) can be calculated during each interval by solving equation (1). The value of φ can be fixed without loss of generality.

To further understand the contribution of a contaminated environment to transmission, the shedding rate φ is chosen such that $\int_t^{t+1} E(t) dt = 1$ for one infectious individual during one day in a clean environment. This shedding rate then depends on the decay rate μ as follow:

$$\varphi(\mu) = \frac{\mu^2}{-1 + e^{-\mu} + \mu} \quad (2)$$

For an interval Δt , the probability to become infected for each of the recipient cats is:

$$p(\mu, \beta) = 1 - e^{-\beta \frac{\int_0^{\Delta t} E(t) dt}{N}} \quad (2)$$

This model assumes shedding to the environment with constant rate parameter φ and decay rate parameter (inactivation of the virus) μ . The transmission occurs with transmission rate parameter β . The parameters μ and β can be estimated by maximum likelihood either directly by optimizing the log likelihood as we did here (Supplementary Material 2) or by Generalized Linear Models using an offset (29). The parameter φ can be then calculated using the estimated value of μ .

The total contribution of both environment and direct contact to transmission from the first day of shedding of one infectious individual is then $\frac{\varphi}{\mu} \beta$ and the basic reproduction number is:

$$R_0 = \frac{\varphi}{\mu} \beta * T \quad (3)$$

where the T is the infectious period. Because for the first day of exposure (donor – recipient) $\int_t^{t+1} E(t) dt = 1$, the transmission rate during this day is equal to β and it represents the expected transmission due to direct contact alone. The reproduction number R due to direct contact alone would be $R = \beta * T$. The apparent contribution of the environment to the total transmission is then the difference between R_0 and R .

Minor Comments:

1. Results section page 4. The experimental designed outlined in this section is very difficult to understand. Understanding this section required re-reading both the results, methods section, and figure legend multiple times. This section should be revised for clarity to clearly explain the experimental design and should also clearly indicate the number of animals and replicates.

Response of authors: We acknowledge that the study design is complex, and that it can be described in the results section as well to enhance readability.

We added a brief description of the study design to the results section:

The study design is explained in detail in the Methods section and summarized in Figure 1. In brief, a total of 16 cats were divided into four replicate groups (group 1, 2, 3 and 4) consisting of either four male or four female

cats per group. Each group was divided in two subgroups, which were housed in two separate pens A and B: cats X.1 and X.2 were housed together in Pen A from D0 until D6 and in Pen B from D7-23, and cats X.3 and X.4 were housed together in Pen B from D0-6 and in Pen A from D7-23, where X=experimental group. All four cats identified by X.1 were inoculated on D0 and all cats identified by X.2 were housed together with X.1 except of on the day of inoculation of cats X.1 (D0). On D6, cats X.1 and X.2 were removed from Pen A and cats X.3 and X.4 were placed in (contaminated) Pen A; while cats X.1 and X.2 were then placed in (clean) Pen B in which cats X.3 and X.4 had been housed until then. Thus, four independently housed pairs of cats were used to assess direct transmission (X.1 and X.2) and contamination of the environment (pen) where new naïve cats (two per contaminated pen: X.3 and X.4) were introduced following the removal of the pair of cats used to assess direct transmission. The sample size was calculated based on published data (10). Of the direct transmission pairs, the four donor cats became infected following inoculation, and three of those transmitted infection to their corresponding contact cats. Of the eight cats exposed to the contaminated pens, only one got infected. Below, we provide the detailed results of the observed infection characteristics and the quantitative assessment of transmission.
(...)

2. In the section on clinical signs and pathology, the authors need to indicate if they expected pathology on day 23. If so, then please provide a rationale and/or reference. If not, please indicate pathology would not be expected at day 23.

Response of authors: No pathology was expected on day 23 and we added this to the results section for clarification.

Clinical signs and pathology

Two of the four inoculated cats occasionally showed mild serous nasal discharge. All 12 cats that had direct contact or indirect contact with the inoculated cat remained without clinical signs. The body weights of all 16 cats remained constant. A video-based analysis of activity of the direct transmission pairs revealed no changes in activity post inoculation compared to before inoculation (baseline measurement), indicating that inoculated and direct contact cats displayed a similar activity pattern regardless of SARS-CoV-2 inoculation/direct exposure (Supplemental Material 1). One cat from Group 4 (Cat 4.3) died on D18 and was necropsied the same day. It was confirmed that the death was unrelated to SARS-CoV-2 infection. All other cats were euthanized on D23. As expected, no lesions were observed in gross pathology, and there were no substantial differences in relative lung weights between the animals upon necropsy (data not shown). Lung tissue showed no SARS-CoV-2-related histopathology. However, mild lung changes - such as lymphoplasmacytic bronchoadenitis, bronchus-associated lymphoid tissue (BALT) hyperplasia, infiltrates of macrophages and few neutrophils in alveolar lumina - were observed in inoculated animals, and in direct and indirect contact animals (data not shown) suggesting these changes were nonspecific. Other evaluated organs (nasal conchae, trachea, tracheobronchial lymph node, duodenum, ileum, colon, mesenteric lymph node and pancreas) showed also no substantial histopathological findings. No viral antigen could be detected by immunohistochemistry in any of the investigated tissues (data not shown).
(...)

3. In general, the figures in panel 2 are difficult to interpret. It would be easier if in panel B the figure were divided to show titers in the transmission pairs separately from the environmental transmission animals.

Response of authors: We changed figure 2b in such a way that only samples collected from the direct and indirect contact animals are displayed. We agree that this modification increases the readability of the figure.

We modified the figure legend as following:

Figure 2. E-gene and subgenomic PCR results on swabs. For total E-gene PCR, the mean log₁₀ RNA copies/mL from two technical replicates of the same swab are shown. The replicates were generated by isolating RNA from the swab sample in duplicate and subsequent PCR. The sgPCR was performed once with the RNA sample from the replicate that showed a lower Ct value in the total E-gene PCR. Shapes indicating the results are jittered so that overlapping shapes can still be observed. The red line indicates the day of removal of cats 1 and 2 from Pen A and housing them in Pen B. **(a)** Swabs from the direct transmission pairs. **(b)** Samples on the left of the red vertical line are collected from Cats 3 and 4 (indirect transmission cats) before environmental exposure housed in Pen B until D6, while samples on the right of the red vertical line are collected from Cats 3 and 4 housed in contaminated Pen A from D6 onwards.

4. Figure 3e and f. Please clarify what the box plots and the data points indicate.

Response of authors: We agree that the figure legends was not sufficiently well explained. We added the following sentences for clarification in the figure legend of figure 3:

(...) **(d)** Overall viral RNA levels of the environment per group and sample type. Viral RNA was detectable throughout the entire study period. **(e)** Distribution of viral RNA load in the different groups. This graph shows the viral E-gene RNA copy numbers (log₁₀) of samples collected in the four different experimental groups (Group 1, 2, 3 or 4), independent of study day and independent of type of sample. **(f)** Distribution of viral RNA load per sample type. This graph shows the viral E-gene RNA copy numbers (log₁₀) of samples collected from the different sample locations (floor, food dish, litter box, toy or wall), independent of study day and independent of experimental group.

5. Discussion, last line of the first paragraph. Please revise. The authors do not demonstrate sustained transmission as they only evaluated a single transmission event in 4 pairs of ferrets. To demonstrate sustained transmission and chain of transmission is required.

Response of authors: According to epidemiological theory, sustained transmission of a pathogen is explained by the R₀ estimate. When R₀ > 1, one can expect sustained transmission and an epidemic to take place; while when R₀ < 1, one could expect that transmission won't be sustained and infection will die out. The statistical analysis and the repetitions in the experiment allowed this assessment (see also Velthuis et al, 2002). We acknowledge that the wording of 'sustained transmission' can be interpreted as 'sustained in a population'. We therefore changed the appropriate sections in the manuscript for clarification.

In the abstract:

These data indicate that transmission between cats is efficient and can be sustained (R₀>1), however, the infectiousness of a contaminated environment decays rapidly (mean duration of infectiousness 1/2.73 days). Despite this, infections of cats via exposure to a SARS-CoV-2-contaminated environment cannot be discounted if cats are exposed shortly after contamination.

In the discussion section:

This study explored direct and indirect transmission of SARS-CoV-2, lineage B.1.22, among domestic cats in an experimental setting. We particularly quantified the duration of infectiousness of an environment with contaminated surfaces. We found that the infectiousness of contaminated surfaces would decay within 8.8 (95%CI: 1.5 – 31.2) hours, making transmission via contaminated surfaces alone inefficient, but it cannot be excluded yet. We also provide further confirmation that cat-to-cat transmission is efficient.

(...)

However, when using sgPCR as an indicator for successful transmission, similar R_0 were estimated for datasets 3 and 4, which were 2.5 (95%CI: 0.97 - 5.15) and 2.12 (95%CI: 0.92 - 4.08), respectively (Table 1). These estimates are in agreement with those made by (10) using published data from direct transmission experiments $R_0 = 3.0$ (95% CI: 1.5 – 5.8) or from household infections $R_0 = 2.3$ (95%CI: 1.1 – 4.9) and provide further certainty that cat-to-cat transmission is efficient. In addition, the analysis also indicates that transmission is a result of both close contact interaction between cats and the contamination of the shared environment, with the later having a lower contribution (1/3 of the overall contribution) to the risk of transmission.

(...)

February 7, 2023

Dr. Nora M. Gerhards
Wageningen Bioveterinary Research
Epidemiology, Bioinformatics & animal models
Houtribweg
39
Lelystad, Flevoland 8221 RA
Netherlands

Re: Spectrum02553-22R1 (Efficient direct and limited environmental transmission of SARS-CoV-2 lineage B.1.22 in domestic cats)

Dear Dr. Nora M. Gerhards:

We appreciate your consideration of the reviewers' comments and for the submission of a revised version addressing the issues raised. As you will see from the referees' comments that additional information needs to be provided. Please fully address the comment from reviewer 2, particularly point 4 in regards to length of infectivity. We ask that this be provided, before we consider you manuscript further.

Link Not Available

Sincerely,

Rafael A. Medina

Journals Department
Reviewer comments:

Reviewer #1 (Comments for the Author):

All comments were addressed

Reviewer #2 (Comments for the Author):

Gerhards et al. have submitted a revised version of their manuscript describing SARS-CoV-2 contact and environmental transmission in cats. The manuscript is much clearer and is substantially improved.

Major Comments:

1. None.

Minor Comments:

1. Lines 125-128. The authors indicate that no histopathological changes were observed in organs collected on day 23. It is important to note and indicate that this does not necessarily mean there were no changes. The animals may have recovered from mild histopathological changes by this time.
2. Figure 3c. In the figure legend please indicate that time point at which PBMCs were collected and used for analyses.
3. Lines 171-175 and Fig 3d-f. It is unclear from the data in the figure how the author's arrived at the conclusion that Group 4 had the highest viral titers. In looking at Fig 3d, a trend may be evident, and it appears that the authors attempted to quantify this trend in Fig 3e and f. However, the data in Fig 3e and f shows box plots at $\sim 10^3$ copies/ml but all the data points (with the exception of Group 3) are at baseline. Thus, it is unclear how the box plots were derived and how the authors arrived at their conclusion. Please revise.
4. Lines 183-185. This reviewer disagrees with the assumption that animals are infectious for the entire duration when they are PCR positive. Except for immunocompromised patients, viral shed does not typically last more than 7-10 days. This becomes more confusing when in the describing the results of the statistical analyses the authors indicate they considered several scenarios and it appears that they do not use a period of 15-23 days in these calculations. If an infectious period of 15-23 days is used, then these calculations are likely an overestimate. Please revise accordingly.
5. Line 428. I think the authors are referring to the dust clothes used to sample the environment when indicating "Environmental tissues". If so, please revise to indicate dust tissues. Alternatively, please clarify what is meant by "Environmental tissues".

Staff Comments:

Preparing Revision Guidelines

Please return the manuscript within 60 days; if you cannot complete the modification within this time period, please contact me. If you do not wish to modify the manuscript and prefer to submit it to another journal, please notify me of your decision immediately so that the manuscript may be formally withdrawn from consideration by Microbiology Spectrum.

Spectrum02553-22 (Efficient direct and limited environmental transmission of SARS-CoV-2 lineage B.1.22 in domestic cats)

Response to reviewers

Reviewer comments:

Reviewer #1 (Comments for the Author):

All comments were addressed

Response of authors: We thank the reviewer for his/her contribution to improve the manuscript.

Reviewer #2 (Comments for the Author):

Gerhards et al. have submitted a revised version of their manuscript describing SARS-CoV-2 contact and environmental transmission in cats. The manuscript is much clearer and is substantially improved.

Major Comments:

1. None.

Minor Comments:

1. Lines 125-128. The authors indicate that no histopathological changes were observed in organs collected on day 23. It is important to note and indicate that this does not necessarily mean there were no changes. The animals may have recovered from mild histopathological changes by this time.

Response of authors: We agree to this comment and specified this in the results section accordingly:

Clinical signs and pathology

(...) Upon necropsy, As expected, no lesions were observed in gross pathology, and there were no substantial differences in relative lung weights between the animals, as expected two to three weeks post SARS-CoV-2 exposure upon necropsy (data not shown). Lung tissue showed no SARS-CoV-2-related histopathology. However, mild lung changes - such as lymphoplasmacytic bronchoadenitis, bronchus-associated lymphoid tissue (BALT) hyperplasia, infiltrates of macrophages and few neutrophils in alveolar lumina - were observed in inoculated animals, and in direct and indirect contact animals (data not shown) suggesting these changes were nonspecific. Yet, transient lung pathology at earlier time points that were resolved until day of necropsy, cannot be excluded. Other evaluated organs (nasal conchae, trachea, tracheobronchial lymph node, duodenum, ileum, colon, mesenterial lymph node and pancreas) showed also no substantial histopathological findings. No viral antigen could be detected by immunohistochemistry in any of the investigated tissues (data not shown). (...)

2. Figure 3c. In the figure legend please indicate that time point at which PBMCs were collected and used for analyses.

Response of authors: We added the required information to the figure legend.

Figure 3. Serology, immunology and environmental contamination. (...) (c) Interferon- γ response of PBMCs, collected on D23 upon necropsy, after stimulation for 48h with spike protein (SP), nucleoprotein (NP), a peptide pool (PEPTIDES), and Concanavalin A (ConA) and medium (Mem) as control, measured by ELISpot analysis. An interferon- γ response was observed in all inoculated cats and

direct contact cats in group 1 and 2 after stimulation with SP, NP, the peptide pool, and ConA. No interferon- γ response was detected in the direct contact cat of Group 4 (indicated by half-filled bullets) or in the indirect contact cats, except of for the positive control ConA. Cats 3.2 and 4.3 were not tested due to an insufficient number of PBMCs. (...)

3. Lines 171-175 and Fig 3d-f. It is unclear from the data in the figure how the author's arrived at the conclusion that Group 4 had the highest viral titers. In looking at Fig 3d, a trend may be evident, and it appears that the authors attempted to quantify this trend in Fig 3e and f. However, the data in Fig 3e and f shows box plots at $\sim 10^3$ copies/ml but all the data points (with the exception of Group 3) are at baseline. Thus, it is unclear how the box plots were derived and how the authors arrived at their conclusion. Please revise.

Response of authors: In figure 3d only the positive samples are shown as bars, and negative samples are indicated by the absence of a bar. Figure 3e summarizes the data from figure 3d per group and 3f per sample type to demonstrate the distribution of viral load. In those two figures e and f, the negative samples are indeed indicated as data points at ~ 0 . The median values were compared to determine which group and which sample types had the highest viral load. We changed figure legend 3 as following:

Figure 3. Serology, immunology and environmental contamination. (...) (d - f) Environmental samples analyzed by E-gene PCR. (d) Overall Viral RNA levels of positive environmental samples per group and sample type. Viral RNA was detectable throughout the entire study period. (e) Distribution of viral RNA load in the different groups. This graph shows the distribution of the viral E-gene RNA copy numbers (log10) of samples collected in the four different experimental groups (Group 1, 2, 3 or 4), independent of study day and independent of type of sample. (f) Distribution of viral RNA load per sample type. This graph shows the distribution of viral E-gene RNA copy numbers (log10) of samples collected from the different sample locations (floor, food dish, litter box, toy or wall), independent of study day and independent of experimental group.

4. Lines 183-185. This reviewer disagrees with the assumption that animals are infectious for the entire duration when they are PCR positive. Except for immunocompromised patients, viral shed does not typically last more than 7-10 days. This becomes more confusing when in the describing the results of the statistical analyses the authors indicate they considered several scenarios and it appears that they do not use a period of 15-23 days in these calculations. If an infectious period of 15-23 days is used, then these calculations are likely an overestimate. Please revise accordingly.

Response of authors: The duration of the infectious period is mostly relevant for the quantification of R and here we considered it to be 6 days as reported in Table 1. The long PCR positive results are not influential in the estimation of the transmission rate parameter since most of the direct transmission (3/4 contact cats) took place within the first 7 days (Figure 2). In practice this is only cat 03 in group 01 which is (with this definition) considered to be infectious for 7 days. In group 03 the contact cat 02 does not become infected but analysis is only done for the direct transmission period which is the first 7 days. In groups 01 and 02 for the direct transmission period the contact cats are found infected just after being moved to a clean environment and infection is counted on the last day of the direct transmission period. Note that for the analysis we only look here at the samples of the cats not of the environment. To further specify this in the manuscript, we modified the results section as following:

Statistical analysis of transmission

For transmission to be considered successful we defined contact cats as being infected and infectious either based on the sgPCR or on seroconversion. For quantification of the transmission rate and the decay rate parameters, the moment the donor and infected contact cats were identified as infected (moment of infection) and infectious was based on the E-gene PCR (evidence of exposure) because of the higher sensitivity of this PCR, i.e. the moment of infection was considered to be 1 - 2 days before the first E-gene-positive sample (latent period). For the

analysis, they were included as possible source of infection from the first day they were observed as E-gene-PCR positive to the time their contact animal became infected (transmission took place). When transmission did not take place, a donor cat was considered infectious for the duration of observed E-gene-positive samples. However, in practice in these experiments the longest infectious period in the analysis used, based on these rules, was 7 days. They were then counted as infectious from the first day until the last day that they were observed as E-gene-PCR positive in any of the samples tested.

5. Line 428. I think the authors are referring to the dust clothes used to sample the environment when indicating "Environmental tissues". If so, please revise to indicate dust tissues. Alternatively, please clarify what is meant by "Environmental tissues".

Response of authors: Indeed, we refer to dust clothes, we specified this in the methods section:

Organ suspensions, environmental samples, and swabs

(...) Environmental ~~tissues~~ dust clothes were collected in individual seal bags and frozen directly at -70°C. To recover viral RNA, 10mL of tissue culture medium (MEM) was added to the tissues and squiggled vigorously, before taking a sample of 85µL in 255µL Trizol-LS for RNA isolation.

Following sampling, all swabs were directly submerged in approximately 2mL of tissue culture medium (MEM), kept on melting ice before vigorously vortexing for 30s on a vortex (Labdancer, VWR International B.V., Amsterdam, the Netherlands), followed by centrifugation for 5 minutes at 1500×g and 4°C. One sample aliquot of 200µL was mixed with 200µL lysis buffer (Molgen, Veenendaal, the Netherlands), and stored at -15°C until RNA isolation.

Additional comment by authors: We furthermore modified a section in the supplemental materials and methods which we had missed during the first round of revision. Therefore, we uploaded a new version of the supplemental material.

April 17, 2023

Dr. Nora M. Gerhards
Wageningen Bioveterinary Research
Epidemiology, Bioinformatics & animal models
Houtribweg
39
Lelystad, Flevoland 8221 RA
Netherlands

Re: Spectrum02553-22R2 (Efficient direct and limited environmental transmission of SARS-CoV-2 lineage B.1.22 in domestic cats)

Dear Dr. Nora M. Gerhards:

We appreciate your consideration of the reviewers' comments and for the submission of a revised version.

Your manuscript has been accepted, and I am forwarding it to the ASM Journals Department for publication. You will be notified when your proofs are ready to be viewed.

Sincerely,

Rafael A. Medina
Editor, Microbiology Spectrum
